# GmWAK1, Novel Wall-Associated Protein Kinase, Positively Regulates Response of Soybean to *Phytophthora sojae* Infection

**DOI:** 10.3390/ijms24010798

**Published:** 2023-01-02

**Authors:** Ming Zhao, Ninghui Li, Simei Chen, Junjiang Wu, Shengfu He, Yuxin Zhao, Xiran Wang, Xiaoyu Chen, Chuanzhong Zhang, Xin Fang, Yan Sun, Bo Song, Shanshan Liu, Yaguang Liu, Pengfei Xu, Shuzhen Zhang

**Affiliations:** 1Soybean Research Institute of Northeast Agricultural University/Key Laboratory of Soybean Biology of Chinese Education Ministry, Harbin 150030, China; 2Soybean Research Institute of Heilongjiang Academy of Agricultural Sciences/Key Laboratory of Soybean Cultivation of Ministry of Agriculture, Harbin 150030, China; 3Key Laboratory of Soybean Molecular Design Breeding, Northeast Institute of Geography and Agroecology, Chinese Academy of Sciences, Harbin 150030, China

**Keywords:** *Glycine max*, wall-associated protein kinase, annexin, pathogenesis-related genes, SA, ROS

## Abstract

Phytophthora root rot is a destructive soybean disease worldwide, which is caused by the oomycete pathogen *Phytophthora sojae* (*P. sojae*). Wall-associated protein kinase (*WAK*) genes, a family of the receptor-like protein kinase (*RLK*) genes, play important roles in the plant signaling pathways that regulate stress responses and pathogen resistance. In our study, we found a putative Glycine max wall-associated protein kinase, GmWAK1, which we identified by soybean GmLHP1 RNA-sequencing. The expression of *GmWAK1* was significantly increased by *P. sojae* and salicylic acid (SA). Overexpression of GmWAK1 in soybean significantly improved resistance to *P. sojae*, and the levels of phenylalanine ammonia-lyase (PAL), SA, and SA-biosynthesis-related genes were markedly higher than in the wild-type (WT) soybean. The activities of enzymatic superoxide dismutase (SOD) and peroxidase (POD) antioxidants in *GmWAK1*-overexpressing (OE) plants were significantly higher than those in in WT plants treated with *P. sojae;* reactive oxygen species (ROS) and hydrogen peroxide (H_2_O_2_) accumulation was considerably lower in GmWAK1-OE after *P. sojae* infection. GmWAK1 interacted with annexin-like protein RJ, GmANNRJ4, which improved resistance to *P. sojae* and increased intracellular free-calcium accumulation. In *GmANNRJ4-OE* transgenic soybean, the calmodulin-dependent kinase gene *GmMPK6* and several pathogenesis-related (*PR*) genes were constitutively activated. Collectively, these results indicated that *GmWAK1* interacts with GmANNRJ4, and *GmWAK1* plays a positive role in soybean resistance to *P. sojae* via a process that might be dependent on SA and involved in alleviating damage caused by oxidative stress.

## 1. Introduction

Phytophthora root rot is a destructive disease in soybean worldwide, which is caused by the oomycete pathogen *Phytophthora sojae* (*P. sojae*) [1,2]. Its incidence is rapid, it spreads widely, and it can thus result in serious economic losses [3,4]. To date, using genetic resistance in soybean has been one of the most effective approaches to reduce the losses caused by *P. sojae.* Soybean cultivars ‘Dongnong 50’ and ‘Suinong 10’ are resistant and susceptible, respectively, to the predominant race 1 of *P. sojae* in Heilongjiang, China [5]; these materials are valuable for the study of plant resistance to *P. sojae.*

Plants have evolved numerous sensor proteins that perceive environmental cues and enable plants to appropriately respond to environmental stresses and pathogens [6]. Understanding how sensor proteins are used to monitor and respond to external signals in these processes is crucial for improving plant resistance to disease. Transmembrane receptor-like protein kinases (RLKs) play fundamental roles in cell–cell and plant–environment communication [7,8]. In plants, the first RLK was identified in *Zea mays* L. [7], and many others have been characterized from a variety of species [9,10,11,12]. RLKs are involved in developmental processes as well as responses to both abiotic and biotic stresses, particularly in plant–pathogen interactions [13,14,15,16].

Cell WAKs are a unique subfamily of plant RLKs that constitute the WAK-RLKs with the RLKs, which is a kinase structural domain responsible for activating the cytoplasmic signaling cascade [17,18]. Cell WAKs have a typical eukaryotic Ser/Thr kinase domain and an extracytoplasmic domain (ectodomain) containing several epidermal growth factor (EGF)-like repeats, which are the hallmark of the WAK subfamily [19,20,21]. Because of the unique structural features of WAK-RLKs, they can transfer signals from the cell wall to the cytoplasm well, whereas the extracellular structural domain can sense the stimulation of the cell wall and enable the transmission of signals via the cytoplasmic kinase structural domain [22,23,24].

WAKs are also involved in responses to pathogens [25,26,27]. In Arabidopsis, salicylic acid (SA) can induce the expression of *AtWAK1*; this induction requires the nonexpression of pathogenesis-related genes (NPR1) [24,25]. Moreover, *AtWAK1* can be induced by methyl jasmonate and ethylene, which are related to fungal pathogens and defense-related signaling [26,27]. In addition to Arabidopsis, WAKs play a key role in the response to pathogens in other species [24], including tomato [28,29], wheat [30], and rice [31,32].

WAK proteins have been implicated in plant responses to minerals [33,34,35,36]. For example, WAKL4 plays an important role in Zn^2+^ accumulation in Arabidopsis shoots [35]. In terms of WAK function, these proteins appear to act as signal messengers in plant cells [23,33]. Via its extracellular domain, AtWAK1 acts as an oligogalacturonide receptor, which interacts with cell-wall-localized molecules [36,37,38]. Taken together, these results suggest a role of WAKs in signal transduction between the extracellular matrix and the cytoplasm during stress responses and development. However, the mechanisms of the WAK-mediated signal transduction pathway and the defense responses regulated by WAK are still unclear.

In our previous study, we found that GmLHP1 (like heterochromatin protein 1) is a negative regulator of the response to infection by *P. sojae* [39]. Through the transcriptome sequencing of soybean seedlings overexpressing GmLHP1, we detected notably enhanced expression of the gene encoding cell-wall-associated receptor protein kinase, designated *GmWAK1* (GenBank accession no. XM_003534738), in the WAK-RLK family. As such, our aims in the current study included (a) using expression analysis and the phenotype to verify whether GmWAK1 is involved in the response to *P. sojae* infection; (b) finding the protein that interacts with GmWAK1 through a yeast two-hybrid assay and verifying its functionality; and (c) examining the physiological indicators and SA content to determine the molecular mechanism through which GmWAK1 responds to *P. sojae*. Our results provide a new perspective for in-depth analysis of the soybean response to *P. sojae* infection.

## 2. Results

### 2.1. GmWAK1 May Be Involved in Response to P. sojae Infection

Through RNA-sequencing (RNA-seq) analysis of the transcriptomes of wild-type (WT) and *GmLHP1*-overexpressing (OE) transgenic soybean lines, we found that the expression of a putative soybean wall-associated protein kinase gene, *GmWAK1* (GenBank accession no. XM_003534738), significantly increased. However, GmLHP1 could not directly bind to the promoter region of *GmWAK1* according to the results of the ChIP-qRT-PCR detection of *GmLHP1*-overexpressing and the WT plants (Appendix A). The expression of *GmWAK1* was significantly higher in *GmLHP1*-overexpressing (OE) transgenic soybean plants than in the WT plants, and the expression of *GmWAK1* in *GmLHP1*-RNAi transgenic soybean lines was markedly lower than that in the WT plants (Figure 1A). These results suggested that GmLHP1 can indirectly regulate the expression of GmWAK1.

### 2.2. Bioinformatics Analysis of GmWAK1

The full length of GmWAK1 is 2130 bp, which encodes a polypeptide of 709 amino acids. The predicted structure of GmWAK1 includes a conserved 111-residue GUB_WAK domain at amino acids 43–187 and a 265-residue PKc-like domain at amino acids 392–657 (Appendix A). To find closely related proteins that might provide clues regarding the function of GmWAK1, we used a neighbor-joining algorithm to construct a phylogenetic tree based on the amino acid sequences of GmWAK1 and the 30 other members of the WAK family. The results of our comparison of GmWAK1 with the two neighboring WAKs with highest homology showed that it shares 68.76% amino acid identity with GmWAK8 (from *Glycine max*) (XP_003534785.2) and 68.22% amino acid identity with GsWAK9 (from *Glycine soja*) (KHN44651.1) (Appendix A). The conserved 265 aa PKc-like domain of GmWAK1 is 96.64% identical to that of GmWAK8 and 93.62% identical to that of GsWAK9 (Appendix A).

### 2.3. GmWAK1 Transcript Levels under Different Treatments

We evaluated the effect of biotic and abiotic stresses on *GmWAK1* mRNA levels in ‘Suinong10’ and ‘Dongnong50’ by quantitative reverse transcription polymerase chain reaction (qRT-PCR). *GmWAK1* was constitutively and most highly expressed in the roots, followed by the leaves and stems; the expression levels of GmWAK1 in the tissues of resistant soybean ‘Suinong 10’ were much higher than those in the tissues of susceptible soybean ‘Dongnong 50’ (Figure 1B). Inoculation of the resistant cultivar ‘Suinong 10’ with *P. sojae* caused a rapid increase in the levels of *GmWAK1* mRNA, which reached a maximum at 48 h after inoculation, followed by a decline by 72 h. In contrast, inoculation with *P. sojae* resulted in no significant change in *GmWAK1* mRNA levels in the susceptible cultivar ‘Dongnong 50’ (Figure 1C). *GmWAK1* expression was also induced by SA, and the SA treatment caused the most rapid increase in *GmWAK1* mRNA levels, which reached a maximum at 9 h with a subsequent rapid decline in expression (Figure 1D). These results suggested that GmWAK1 is primarily involved in the response to *P. sojae* and SA treatment.

### 2.4. Subcellular Localization of the GmWAK1

To verify the subcellular location of GmWAK1, we expressed a GmWAK1-GFP fusion protein in Arabidopsis protoplasts. We observed the green fluorescent protein (GFP) signal at the cell surface of protoplasts harboring GmWAK1-GFP; and marker protein P2300-35S-PIP2-mcherry in transgenic protoplasts showed a red fluorescence on the cell membrane (Figure 2A), similar to what was observed for the pathogenesis-related protein GmSnRK1 (Figure 2B) [40]. We observed the GFP signal in the entire cell of protoplasts transformed with 35S:GFP plasmids (Figure 2C). These results indicated that GmWAK1 is a membrane-localized protein.

### 2.5. GmWAK1 Is Positive Regulator in Soybean Resistance to P. sojae

To study the role of *GmWAK1* in response to *P. sojae*, we generated *GmWAK1*-OE and *GmWAK1*-RNAi transgenic soybean lines (Appendix A). To test the *P. sojae* resistance of the T_2_ transgenic lines, we applied the zoospores of *P. sojae* to their cotyledons and leaves. After 3 days of incubation, we observed notable differences among the soybean lines. In the *GmWAK1*-RNAi lines, the cotyledons inoculated with *P. sojae* exhibited water-soaked lesions and were softer than those of WT; we observed almost no disease symptoms in the *GmWAK1*-OE lines. The *P. sojae* lesions on the *GmWAK1*-OE plants were significantly smaller than those on WT plants (*p* < 0.01), but larger than those on the *GmWAK1*-RNAi plants (Figure 3A). We obtained similar results after the inoculation of true leaves with *P. sojae*. After 3 days of incubation, the true leaves of WT and *GmWAK1*-RNAi transgenic soybean plants exhibited watery and rotting lesions, whereas those of the *GmWAK1*-OE transgenic soybean plants remained healthy (Figure 3B). As observed for the cotyledons, the lesion areas were significantly smaller in the *GmWAK1*-OE transgenic soybean plants and significantly larger in the *GmWAK1*-RNAi transgenic soybean plants relative to the WT plants (Figure 3C). The relative expression of *GmTEF1* was significantly lower in in cotyledons and true leaves of *GmWAK1*-OE lines than in the WT plants but significantly higher in the *GmWAK1*-RNAi lines (*p* < 0.01) (Figure 3D,E). These results suggested that when *GmWAK1* is overexpressed, the transgenic soybean plants acquire higher resistance to *P. sojae* infection.

### 2.6. GmWAK1-Dependent SA Signaling in Response to P. sojae

To analyze whether the GmWAK1-regulated defense response was dependent on these phytohormones, we measured the SA content in the hairy roots of both GmWAK1 transgenic and WT plants. As shown in Figure 4A, the SA levels were significantly higher in GmWAK1-OE transgenic hairy roots than in the EV, and those of GmWAK1-RNAi were lower than those in the WT plants. The results showed that the phenylalanine ammonia-lyase (PAL) activity was significantly higher in the hairy roots of GmWAK1-OE transgenic plants than in those of WT plants after infection with *P. sojae* at 24 h postinoculation (hpi). The PAL activity in the hairy roots of GmWAK1-RNAi soybean was lower than that in the hairy roots of WT plants (Figure 4B). The expression of *GmPAL* in GmWAK1-OE was significantly higher than that in the WT plants; conversely, the RNAi of GmWAK1 led to reduced GmPAL activity relative to the control (Figure 4C). All these results showed that GmWAK1 plays an important role in the SA pathway.

### 2.7. Soybean GmWAK1-Regulated Defense Response to P. sojae Involves Alleviating Oxidative Stress Damage

Here, the superoxide dismutase (SOD) and peroxidase (POD) activities were much higher in GmWAK1 transgenic soybean plants than in the WT plants under both the mock treatment and at 24 hpi (Figure 5A,B). The expressions of GmSOD and GmPOD in GmWAK1-OE were significantly higher than those in the WT; the expressions in GmWAK1-RNAi were lower than those in the WT (Appendix A). Furthermore, the accumulation of ROS in GmWAK1-RNAi transgenic soybean plants was much higher than in the WT plants; the overexpression of GmWAK1 in soybean was lower than that in WT (Figure 5C). The accumulation of H_2_O_2_ in GmWAK1-OE transgenic plants was lower than in the WT plants, but it was higher in the transgenic GmWAK1-RNAi plants (Figure 5D). According to the results, we inferred that due to the elevated expression level of the corresponding enzyme genes in GmWAK1 transgenic soybean plants, the antioxidant enzyme activity increased, which may maintain certain lower steady-state levels of ROS_,_ preventing harm from *P. sojae* infection.

### 2.8. GmWAK1 Interacts with GmANNRJ4

We screened the proteins that interact with GmWAK1 with immunoprecipitation-mass spectrometry. When induced with 0.8 mM IPTG at 37 °C for 4 h in LB medium, the GmWAK1-His protein expression levels were the highest. According to the results of SDS-PAGE analysis, the molecular weight of the inducible protein was about 120 kDa (Appendix A). We performed Co-IP assays and LC-MS/MS analysis to identify proteins that interact with GmWAK1 (Appendix A). We identified 12 candidate GmWAK1-interacting proteins (Appendix A). We selected a cDNA corresponding to annexin-like protein RJ4 (GmANNRJ4; LOC100813609) for further study because of its relatively high mass spectrometry score and its previously reported role in stress responses and pathogen resistance [41,42,43,44,45,46,47]. To confirm that these proteins interact in the plants, we used a bimolecular fluorescence complementation (BiFC) assay. We detected notable yellow fluorescence in the cytosol of Arabidopsis protoplasts after cotransformation with both N-terminal yellow fluorescent protein (YFP^N^)-tagged GmWAK1 and C-terminal YFP (YFP^C^)-tagged GmANNRJ4 (Figure 6A). In accordance with these results, the results of the glutathione S-transferase pull-down assay showed that its GmWAK1-his recombinant protein can be pulled down by GmANNRJ4-GST but not by GST alone (Figure 6B), further indicating that GmWAK1 interacts with GmANNRJ4 *i*. These results suggested that GmWAK1 directly interacts with GmANNRJ4.

### 2.9. GmANNRJ4 can Regulate Intracellular Ca^2+^ Concentrations and Promotes GmMPK6 Expression

GmANNRJ4 belongs to the annexin family of proteins, some of which are involved in regulating intracellular Ca^2+^ concentrations [48,49,50]. To determine whether GmANNRJ4 also has this function, we measured the cytoplasmic-free Ca^2+^ concentration in the hairy roots of transgenic transformed soybean with *GmANNRJ4*-overexpression or *GmANNRJ4*-RNAi constructs. The results showed that the concentration of free Ca^2+^ in the hairy roots of plants overexpressing *GmANNRJ4* was higher than that in the hairy roots of WT plants; conversely, the RNAi of *GmANNRJ4* led to a reduced free Ca^2+^ concentration relative to the control (Figure 6C).

To gain further insight into the function of *GmANNRJ4*, we used qPCR to analyze the expression of *GmMPK6* in the hairy roots of *GmANNRJ4*-OE and *GmANNRJ4*-RNAi soybean (Figure 7) relative to that in the WT. The relative expression of *GmMPK6* was significantly higher when *GmANNRJ4* was overexpressed, whereas it was significantly lower in the hairy roots of *GmANNRJ4*-RNAi (Figure 6D). The expression of GmMKK4, which can phosphorylate and activate GmMPK6, was also significantly higher when *GmANNRJ4* was overexpressed, whereas it was significantly lower in the hairy roots of *GmANNRJ4*-RNAi (Appendix A). GmMKK4 and GmMPK6 were also highly expressed in the hairy roots of *GmWAK1*-OE soybean relative to those in the hairy roots of WT plants (Appendix A). This finding suggested that *GmANNRJ4* can positively regulate the expression of *GmMPK6*, and the regulation of *GmMPK6* may be indirectly caused by the regulation of intracellular Ca^2+^ concentration by *GmANNRJ4*.

### 2.10. GmANNRJ4 Enhances Resistance to P. sojae and Positively Regulates Expression of PR Genes in Hairy Roots of Transgenic Soybean

We examined the response of *GmANNRJ4*-OE and *GmANNRJ4*-RNAi in the hairy roots of soybean to *P. sojae* infection. The hairy roots of *GmANNRJ4*-OE transgenic plants displayed significantly enhanced resistance compared with those of the WT plants after 2 days of *P. sojae* infection, and *GmANNRJ4*-RNAi showed the opposite results (Figure 7A). We measure the expression of *PR* genes in the hairy roots of *GmANNRJ4*-OE and *GmANNRJ4*-RNAi soybean by qRT-PCR. The expressions of *GmPR1*, *GmPR*2, and *GmPR10* were significantly higher in the hairy roots of *GmANNRJ4*-OE (Figure 7B) and significantly lower in the hairy roots of *GmANNRJ4*-RNAi (Figure 7C) compared with those in the hairy roots of WT plants. However, *GmPR3* expression was not affected by GmANNRJ4 over- or underexpression (Figure 7B,C). The above results suggest that the expression of *GmANNRJ4* influences the expression of *PR* genes.

## 3. Discussion

*GmLHP1* is an SA-inducible gene that inhibits the expression of GmWRKY40 and negatively regulates plant immunity [39,51]. Moreover, GmLHP1 interacts with GmBTB/POZ, promotes the ubiquitination and degradation of LHP1, and positively regulates the response of soybean to *Phytophthora sojae* infection and the expression of GmBTB/POZ [39,51]. In this study, the *GmWAK1* gene was not directly regulated by *GmLHP1* or the disease-resistance gene *GmWRKY40*, but it was upregulated in both overexpressed transgenic plants. These data suggested that some other genes that upregulate the expression of GmWAK1 and are involved in the regulation of *P. sojae* resistance are regulated by GmLHP1.

The plant cell wall not only maintains cell structure during plant growth and development but also responds to environmental stimuli and microbial pathogens [52]. The cell wall damage caused by pathogens is sensed by specific membrane proteins that transmit the signal across the membrane and activate the defense responses of the cell, thus improving plant pathogen resistance [19]. Among the membrane proteins involved in sensing cell wall damage are wall-associated receptor-like protein kinases (WAKs), which bind to the plant cell wall and have typical structure that consists of an extracellular domain, a transmembrane domain, and an intracellular kinase domain [53]. When plant cell walls are attacked by pathogens, oligogalacturonides (OGAs) are produced and then specifically recognized by AtWAK1 [20]. Activated AtWAK1 transmits this extracellular signal into the cell, which triggers the corresponding intracellular defense responses [38]. GmWAK1 is a membrane-localized protein; this location is consistent with a role in perceiving extracellular signals and transmitting these signals to the inside of the cell [18].

Here, we found that GmWAK1 interacted with GmANNRJ4. GmANNRJ4 is a member of the annexin protein (ANN) family. Annexins in plant membranes have a multisegment alpha-helix ANN domain that specifically binds to Ca^2+^; thus, they are also called Ca^2+^-dependent phospholipid binding proteins [54]. Annexins form permeable calcium channels on cell membranes and play a major role in regulating the concentrations of reactive oxygen species and free Ca^2+^ in plant cells [49,50,55]. Therefore, annexins are thought to directly form a Ca^2+^ flow pathway on the cell membrane and participate in various responses to adverse conditions by regulating Ca^2+^ concentrations and activating calmodulin- and Ca^2+^-dependent proteins [56]. Annexins play an important role in the response to abiotic and biotic stresses [57,58,59,60,61,62,63,64]. Until now, however, whether the *annexin* gene participates in the response to *P. sojae* had not been reported. As a candidate receptor for the transmembrane transmission of a disease-resistance signal, exploring the downstream proteins interacting with GmWAK1 is crucial to understanding its potential mechanism of action. An example is Arabidopsis AtWAK1, the extracellular domain of which binds to pectin ligands in the cell wall; AtWAK1 can activate MPK6-dependent stress responses by binding to the pectin fragments produced by the cell wall after infection by pathogens [21]. GmMKK4 could phosphorylate and activate GmMPK6 to enhance the resistance of soybean to *P. sojae* and promoted the expression of *PR* genes (*GmPR1*, *GmPR2*, *GmPR5*, and *GmPR10*) in response to *P. sojae* infection [65]. In this study, the expression of *GmANNRJ4* in soybean significantly increased when infected by *P. sojae*, indicating that GmANNRJ4 participates in the process of soybean resistance to *P. sojae*. The hairy roots of transgenic soybean overexpressing *GmANNRJ4* showed significantly increased resistance to *P. sojae,* indicating a positive regulation of soybean resistance to *P. sojae*. In determining the relationship between GmWAK1 and GmMPK6, we found that GmWAK1 can also increase the expressions of GmMKK4 and GmMPK6 in transgenic soybean plants overexpressing *GmWAK1* and *GmANNRJ4.* We also found that the expressions of the *PR* genes (*GmPR1*, *GmPR2*, *GmPR5* and *GmPR10*) were increased. Based on previous findings regarding annexin function, we hypothesized that GmWAK1 interacts with GmANNRJ4 and that both GmWAK1 and GmANNRJ4 are involved in regulating soybean resistance to *P. sojae*.

SA mediates systemic acquired resistance (SAR), which can control plant nutrition as well as the production of pathogens and improve disease resistance of organisms [66,67,68,69,70]. Zhang et al. [51] reported that GmBTB/POZ, via a process dependent on SA, plays a positive role in *P. sojae* resistance and the defense response in soybean. GmWAK1 is primarily involved in the response to *P. sojae* and SA treatment. SA accumulation and the transcript abundance of SA biosynthesis genes were much higher in GmWAK1 leaves than in those of WT plants. Herein, we determined that GmWAK1 played an important role in the SA pathway. As with SA, ROS also plays an important role in plant defense [71]. When under stress, a high concentration of H_2_O_2_ can kill pathogens and induce immune responses; low concentrations of H_2_O_2_ can also induce the expression of a series of genes encoding defense response proteins upon pathogen infestation, enhancing their resistance to pathogens [72]. The removal of excess ROS from plants can improve the resistance of plants to many pathogens [72,73,74,75]. Plants have well-developed antioxidant defense systems that efficiently scavenge ROS, involving the antioxidant enzymes SOD and POD [76]. SOD can reduce O^2-^ levels by further dismutation to H_2_O_2_ [77,78]; POD can reduce H_2_O_2_ levels through the transport of electrons [79,80]. Maintaining high SOD and POD activities can result in reductions in H_2_O_2_ and ROS accumulation. In this study, the activities of SOD and POD were significantly increased, and the accumulations of ROS and H_2_O_2_ were kept markedly lower in GmWAK1-OE after *P. sojae* infection. The overexpression of GmWAK1 could enhance the resistance of soybean to *P. sojae*. This implied that the levels of SA and ROS might be critical for GmWAK1 in the response to *P. sojae* infection.

GmWAK1 is a receptor protein kinase associated with the cell wall in soybean. In this study, we examined the relationship between this protein and *Phytophthora sojae* infection. *P. sojae* significantly induced expression of *GmWAK1* in ‘Suinong 10’, which is a soybean cultivar resistant to this pathogen, but only slightly induced the expression in ‘Dongnong 50’, the susceptible soybean cultivar. The SA levels were significantly higher in the hairy roots of transgenic GmWAK1-OE than in those of EV. This suggested that *GmWAK1* may be a gene related to the *P. sojae* pathway in soybean. Transgenic soybean plants overexpressing *GmWAK1* were significantly more resistant to *P. sojae*. Conversely, when the expression of *GmWAK1* was suppressed by RNAi, the resistance to *P. sojae* was reduced. Based on the results, we propose the model depicted in Figure 8 for the promotion of resistance to *P. sojae* by *GmWAK1* in soybean. GmLHP1 indirectly regulates GmWAK1. when soybean is infected by *P. sojae*, the expression of *GmWAK1* increases, the expression of endogenous SA is upregulated, and the activities of SOD and POD are upregulated, which reduce the accumulation of O^2−^ and H_2_O_2_, to be maintained at a certain lower steady-state level of ROS. When GmANNJ4 is overexpressed, calmodulator channels open to allow for the influx of calcium ions, whereas GmWAK1 interacts with GmANNRJ4 and activates the calmodulin-dependent kinase GmMPK6 by regulating intracellular Ca^2+^ concentrations. This ultimately promotes the expression of disease-related *PR* genes (*GmPR1*, *GmPR2*, and *GmPR10*) to enhance the resistance to *P. sojae*. This regulatory network provides a new perspective for in-depth analysis of the response of soybean to *P. sojae* infection and provides a new avenue for molecular breeding of disease resistance in soybean. In conclusion, *GmWAK1* is induced by *P. sojae* infection and plays a positive regulatory role in the response of soybean to this pathogen.

## 4. Materials and Methods

### 4.1. Plant Materials, P. sojae Treatment, and Primers

In this study, we used soybean cultivars ‘Dongnong 50’ and ‘Suinong 10’, which are resistant and susceptible, respectively, to the predominant race of *P. sojae*, race 1, in Heilongjiang, China [5]. We sowed all the seeds in pots in a growth chamber that we set to 25/20 °C (night/day) and 80% relative humidity with a 16/8 h light (110 PAR light intensity)/dark cycle. Fourteen days after planting, we inoculated soybeans at the first-node stage (V1 [81]) with *P. sojae* race 1 following the method described by Ward et al. [82]. We removed unifoliate leaves at 0, 3, 6, 9, 12, 24, 48, and 72 h after inoculation. All primers are listed in Appendix A.

### 4.2. Isolation and Sequence Analysis of GmWAK1

We isolated the total RNA from ‘Suinong 10’ and ‘Dongnong 50’ plants using TRIzol^®^ reagent (Invitrogen, Shanghai, China; part number 15596026.). For reverse transcription, we used an M-MLV reverse-transcriptase kit (Takara, Dalian, China; part number 2641A). We isolated the full-length cDNA sequence of *GmWAK1* from the cDNA of ‘Suinong 10’. We inserted PCR products into a pEASY^®^ Blunt vector (Transgen Biotech, Beijing, China; part number CB101). To obtain the protein sequence data and analyze the nucleotides, we used the NCBI bioinformatics tools (http://www.ncbi.nlm.nih.gov/blast, accessed on 30 July 2022). We analyzed the predicted protein structure with SMART (http://smart.embl-heidelberg.de, accessed on 30 July 2021). We aligned the sequences using DNAMAN software (http://www.lynnon.com, accessed on 30 July 2022). Based on the nucleotide sequences of *GmWAK1* and other *WAK* members, we generated phylogenetic trees using MEGA 5.1 software (http://www.megasoftware.net, accessed on 30 July 2022).

### 4.3. RT-PCR and qRT-PCR Analysis

We performed quantitative real-time PCR (qRT-PCR) to analyze the *GmWAK1* transcript levels using a real-time PCR kit (Toyobo, Japan; part number FSQ-201) [83]. As a housekeeping gene, we used *GmEF1β* (GenBank accession no. NM_001248778) in soybean as an internal reference to normalize all data. We calculated relative transcript levels using the 2^−ΔΔCT^ method. We performed three biological replicates per experiment per line.

### 4.4. Chromatin Immunoprecipitation (ChIP) qPCR Assay

For the ChIP assays, WT, p35S: Flag-*GmLHP1* transgenic plants and p35S: Flag-Myc-*GmWRK40* were subjected to chromatin extraction and immunoprecipitation as described by Saleh et al. [84]. The antibodies we used during the ChIP were anti-MYC-Tag Mouse mAb (Agarose Conjugated) (Abmart, code number M20012) and anti-DYKDDDDK-Tag Mouse mAb (Agarose Conjugated) (Abmart, code number M20018). The immunoprecipitated DNA samples were used as templates for qPCR assays. The primers used for this experiment are listed in Appendix A.

### 4.5. Subcellular Localization of GmWAK1

We cloned the coding sequences (CDs) of *GmWAK1* into the N-terminus of green fluorescent protein (GFP) under the control of the constitutive CaMV35S promoter using the primer pairs GmWAK1-GF and GmWAK1-GR (Appendix A). We transformed the resulting expression vectors, 35S:GmWAK1-GFP and P2300-35S-PIP2-mcherry (RFP), into Arabidopsis protoplasts via polyethylene glycol (PEG)-mediated transfection as described by Yoo et al. [85]. After 16–20 h of incubation at 25 °C and under light, we imaged the GFP/RFP signal with a TCS SP8 confocal spectral microscope imaging system (Leica, Germany).

### 4.6. Inducing the Expression of GmWAK1 Fusion Protein

We constructed the His-tagged pET29b (+) -GmWAK1 recombinant fusion plasmid, which we transformed into *E. coli* BL21 (DE3) cells. We added 0.8 mM isopropyl-b-D-thiogalactoside (IPTG) at 37 °C, which we shook at 220 rpm with a Thermostatic Shaker in LB medium for 4 h. We analyzed the recombinant protein SDS-PAGE.

### 4.7. Vector Construction and Transformation of Soybean

We cloned the CDs of *GmWAK1* into plant expression vector pCAMBIA3301-4Myc with gene-specific primers (Appendix A). We amplified the cDNA fragment of *GmWAK1* using the *GmWAK1* RNAi-F/R primer set, which we inserted into vector pFGC5941 [86] (Appendix A). We individually introduced the recombinant construct *GmWAK1*-OE and *GmWAK1*-RNAi vector into *A. tumefaciens* strain LBA4404, as previously described [87]. For the soybean transformation, we used susceptible cultivar ‘Dongnong 50’ as the explant the generation, following the *Agrobacterium*-mediated transformation method described by Paz et al. [88]. We constructed the plant overexpression vector 35S:*GmANNRJ4* and vector *GmANNRJ4*-RNAi [86] (Appendix A) to obtain transgenic hairy roots according to previously published methods [89,90]. We identified overexpression and RNAi transgenic soybean plants (T_1_) or transgenic hairy roots by PCR amplification, and we used a PAT/bar test strip to identify bar screening markers, which we developed for T_2_ transgenic soybean plants for the following analyses.

### 4.8. Assessment of Pathogen Resistance and the Disease Response

We grew the transgenic soybean seeds under a 16/8 h light/dark photoperiod at 25 °C and 70% relative humidity in a greenhouse. For disease resistance analysis, we treated the living cotyledons of the WT and transgenic soybean plants at the emergence stage (V1) [81] with *P. sojae* zoospores (approximately 1 × 10^5^ spores mL^−1^), following the methods described by Morrison and Thorne [91], and we inoculated the leaves following the procedure described by Dong et al. [92]. To study whether the *GmANNRJ4* conferred resistance to pathogen infection, we performed artificial inoculation procedures as described by Ward et al. [82]. We determined the relative biomass of soybean in the infected root system after 3 d based on the transcript level of the soybean *TEF1* gene, using soybean *GmEF1β* as an internal reference gene, according to Chacón et al. [93] (Appendix A). Each experiment included three replicates, each with three technical replicates.

### 4.9. Bimolecular Fluorescence Complementation (BiFC) Assay

To validate the interactions between GmWAK1 and GmANNRJ4, we performed a BiFC assay based on yellow fluorescence protein (YFP). We separately cloned the coding sequences of *GmWAK1* and *GmANNRJ4* into serial pSAT6 vectors encoding either N- or C-terminal-enhanced yellow fluorescent protein fragments (Appendix A). We used the resulting constructs for the transient assays via PEG transfection of Arabidopsis protoplasts, as described by Yoo et al. [85]. We imaged the transfected cells with a TCS SP8 confocal spectral microscopy imaging system (Leica).

### 4.10. Pull-Down Assays

We cloned *GmWAK1* into the pET29b (+) expression vector and *GmANNRJ4* into the pGEX-4T-1 expression vector. His-GmWAK1 and glutathione S-transferase (GST)-GmANNRJ4 proteins were separately produced in *E. coli* BL21 (DE3) cells, which we then harvested and purified using a GST-Sefinose kit (Sangon, China, code number C590927) or a His-bind Purification Kit (Merck Millipore, Burlington, MA, USA, code number 70239-3). We performed the pull-down assay as described by Yang et al. [94], with minor modifications. In a total volume of 1 mL of GST binding buffer (Sangon), we incubated the GST or GmANNRJ4-GST recombinant proteins for 1 h at 4 °C with 400 mL of GST resin (Sangon), after which we added an equal volume of the GmWAK1-His recombinant protein and then incubated for 6 h at 4 °C. We washed the binding reaction five times with binding buffer, each for 10 min at 4 °C; then, we eluted the pulled-down proteins by boiling, which we separated on a 12% SDS-PAGE gel and immunoblotted with anti-His antibody and anti-GST antibody (Abmart, Berkeley Heights, NJ, USA; part numbers M20020 and M20025).

### 4.11. Determination of Plant Hormone and Antioxidant Enzyme Activity Levels

We measured the SA levels referencing the method described by Pan et al. [95]. We measured SOD and POD enzyme activities according to the method described by Wang et al. [96]. We determined hydrogen peroxide (H_2_O_2_) accumulation according to the method of Velikova et al. [97]. We measured the rate of ROS generation according to the method described by Qian et al. [98]. Each experiment included three replicates, each with three technical replicates.

### 4.12. Intracellular Calcium Concentration Measurement

We individually introduced the *35S:GmANNRJ4*-OE and *35S:GmANNRJ4*-RNAi constructs into *Agrobacterium tumefaciens* strain K599 using the freeze–thaw method [86]. We generated transgenic soybean hairy roots following the methods described by Graham et al. [89] and Kereszt et al. [90] using *A. rhizogenes*-mediated transformation. We sampled the transgenic soybean hairy roots to extract intracellular free calcium, and we used a calcium ion fluorescence probe (Fluo-3 AM) to measure cytoplasmic free calcium concentrations according to the method described by Yu and Hinkle [99].

### 4.13. Statistical Analysis

In this study, we performed all the experiments three times. To analyze all the results and to determine the statistical significance between different measurements, we used Student’s *t* test. We considered a difference statistically significant when * *p* < 0.05 or ** *p* < 0.01.

## 5. Conclusions

In our study, we investigated the biological functions of soybean *GmWAK1* in response to *P. sojae* infection. We found that when soybean was infected by *P. sojae*, the expression of *GmWAK1* increased, the expression of endogenous SA was upregulated, and the activities of SOD/POD were upregulated. GmWAK1 also helped to maintain lower levels of H_2_O_2_ and ROS accumulation. When GmANNJ4 was overexpressed, Calmodulator channels opened to allow for calcium ion influx, whereas GmWAK1 interacted with GmANNRJ4 and activated the calmodulin-dependent kinase GmMPK6 by regulating intracellular Ca^2+^ concentrations. This ultimately promoted the expression of disease-related PR genes (*GmPR1, GmPR2,* and *GmPR10*) to enhance soybean resistance to *P. sojae*. This regulatory network provides a new perspective for in-depth analysis of the response of soybean to *P. sojae* infection and provides a possible new avenue for the molecular breeding of soybean disease resistance. In conclusion, GmWAK1 is induced by *P. sojae* infection and plays a positive regulatory role in the soybean response to this pathogen.

## Figures and Tables

**Figure 1 ijms-24-00798-f001:**
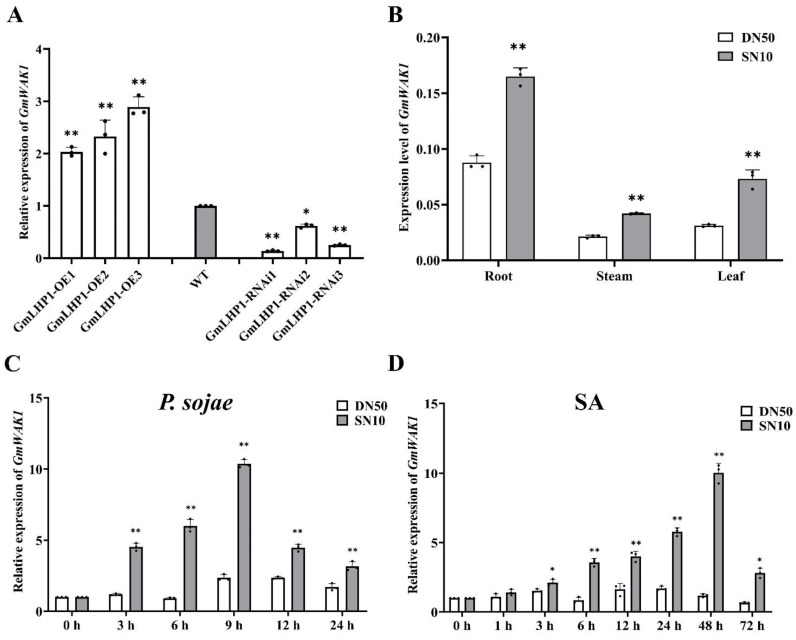
Analysis of expression of *GmWAK1* by quantitative real-time PCR (qRT-PCR)**.** (**A**) Expression of *GmWAK1* genes, *GmLHP1*-overexpressing (OE), RNAi transgenic plants, and WT plants. (**B**) Tissue specificity analysis of ‘Dongnong50’ and ‘Suinong10’ of *GmWAK1* gene. (**C**) Relative *GmWAK1* expression at different times (0, 6, 12, 24, 48, and 72 h) after *Phytophthora sojae* (*P. sojae*) infection of ‘Suinong 10’ and ‘Dongnong 50’. (**D**) Relative *GmWAK1* expression at different times (0, 3, 6, 9, 12, and 24 h) after salicylic acid (SA) (0.5 mM) treatment in ‘Suinong 10’ and ‘Dongnong 50’. Three biological replicates, each with three technical replicates; * *p* < 0.05, ** *p* < 0.01, Student’s *t* test. Error bars indicate standard deviations.

**Figure 2 ijms-24-00798-f002:**
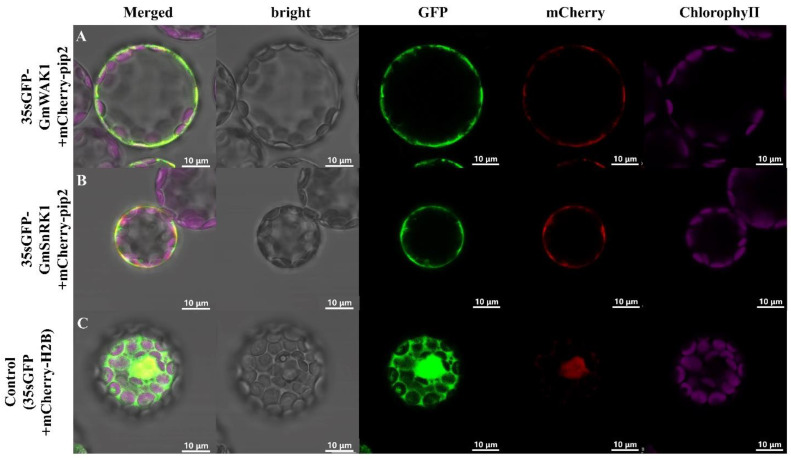
Subcellular localization of GmWAK1. GmWAK1-GFP fusion protein was transiently expressed in Arabidopsis protoplasts; plasmid containing p2300-35S-H2B-mCherry (recombinant histone H2B) was cotransformed as a nuclear marker; p2300-35S-PIP2-mCherry (plasma membrane intrinsic proteins) was cotransformed as a cell membrane marker. Images of bright-field, green fluorescent protein (GFP) fluorescence (green) only, red fluorescent protein (mCherry) fluorescence (red) only, chlorophyll autofluorescence (purple) only, and their combination. Scale bars: 10 µm.

**Figure 3 ijms-24-00798-f003:**
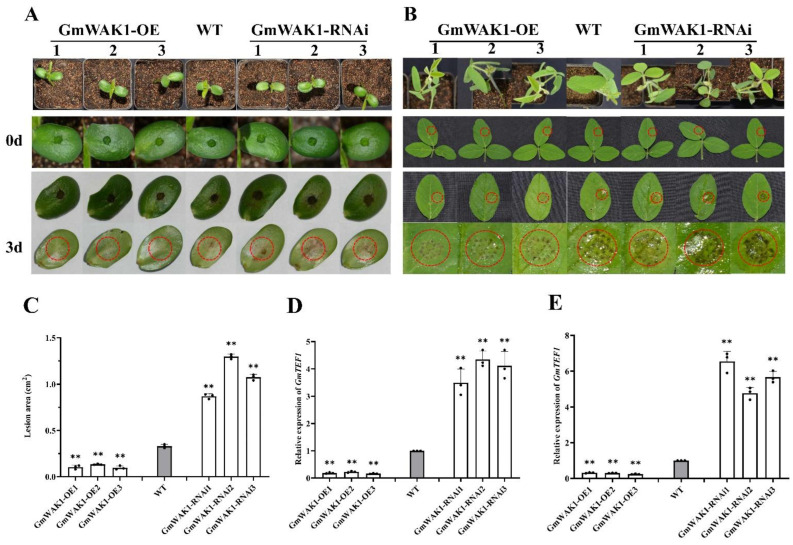
Phenotype identification of GmWAK1 under *P. sojae* influence in transgenic soybean plants. (**A**,**B**) Disease symptoms on living cotyledons and leaves of transgenic and WT plants treated with *P. sojae* inoculum at 0 and 3 days, respectively. (**C**) Lesion area of transgenic lines and WT plants detected after 3 days of incubation with *P. sojae*. (**D**,**E**) Results of qRT-PCR analysis of relative biomass of *P. sojae* in *GmWAK1* transgenic lines and WT soybean based on *P. sojae TEF1* transcript levels. Three biological replicates, each with three technical replicates; ** *p* < 0.01, Student’s *t* test; error bars indicate standard deviation.

**Figure 4 ijms-24-00798-f004:**
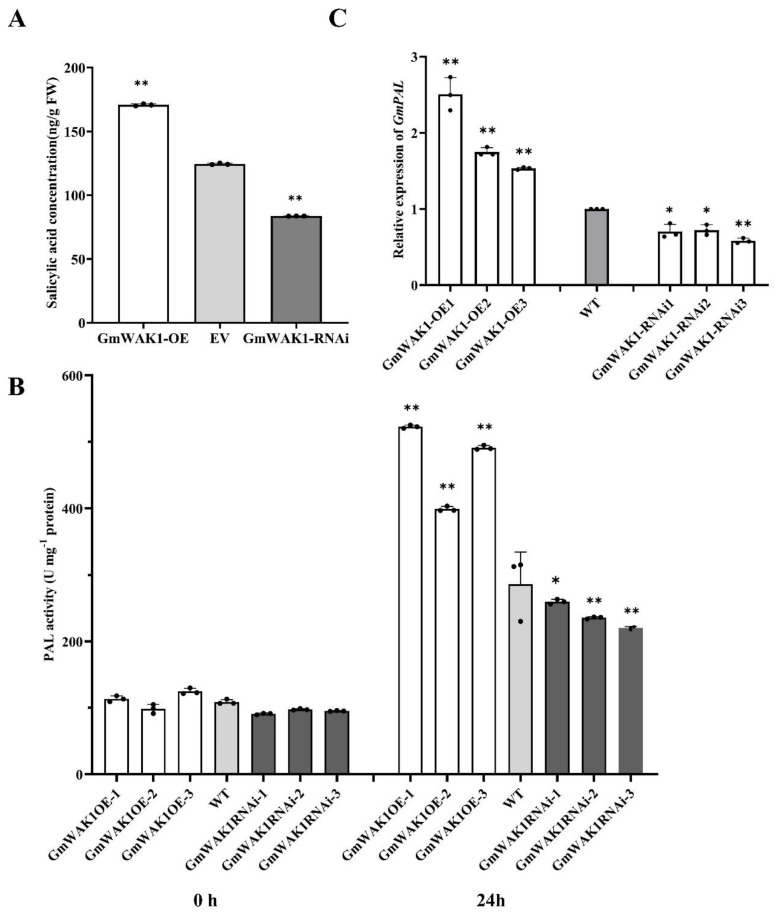
Investigation of relationship between GmWAK1 and salicylic acid (SA). (**A**) SA contents in leaves of transgenic and WT soybean. (**B**) The phenylalanine ammonia-lyase (PAL) activity in transgenic soybean plants and mock-treated WT plants at 0 and 24 h after *P. sojae* infection. (**C**) Relative expression *GmPAL* gene in leaves of transgenic and WT plants. Three biological replicates, each with three technical replicates; * *p* < 0.05, ** *p* < 0.01, Student’s *t* test; error bars indicate standard deviation.

**Figure 5 ijms-24-00798-f005:**
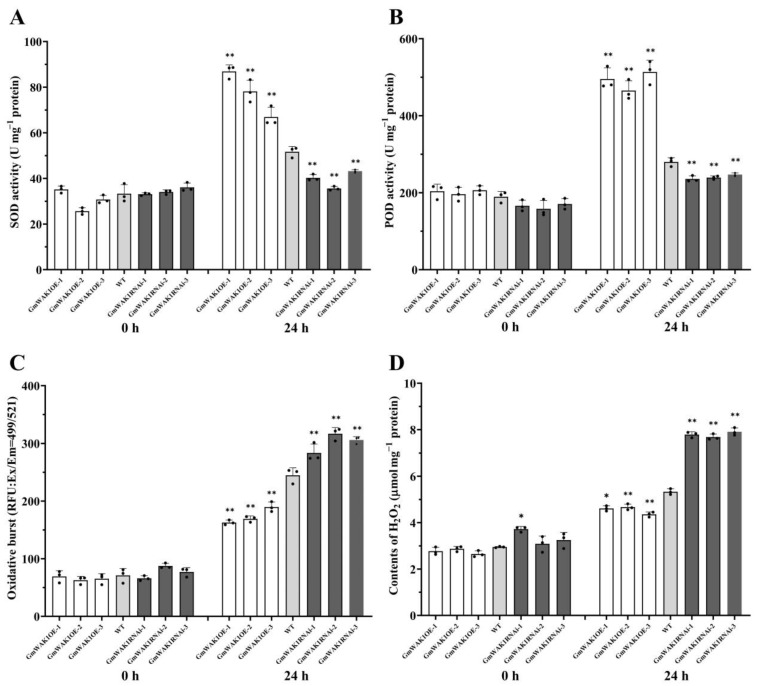
Activities of superoxide dismutase (SOD) (**A**), peroxidase (POD) (**B**), reactive oxygen species (ROS) (**C**), and hydrogen peroxide (H_2_O_2_) (**D**) accumulation. Relative activity of POD and SOD and relative contents of total ROS and H_2_O_2_ in transgenic soybean plants and mock-treated WT plants at 0 and 24 h after *P. sojae* infection. Three biological replicates, each with three technical replicates; * *p* < 0.05, ** *p* < 0.01, Student’s *t* test; error bars indicate standard deviation.

**Figure 6 ijms-24-00798-f006:**
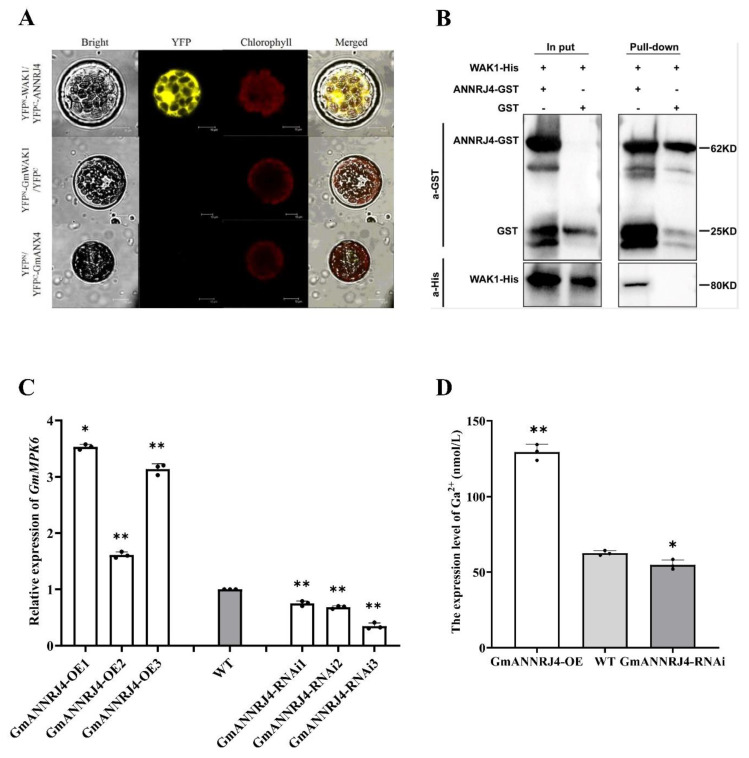
Interaction of GmWAK1 with GmANNRJ4. (**A**) Results of Bimolecular fluorescence complementation (BiFC) analysis of interaction between GmWAK1 and GmANNRJ4 in Arabidopsis protoplast cells. Yellow fluorescent protein (GFP) fluorescence (yellow) was observed by confocal laser microscopy. Scale bars: 10 µm. (**B**) Interaction between GmWAK1 and GmANNRJ4 as revealed by pull-down assay. (**C**) Results of analysis of intracellular Ca^2+^ concentration regulation. Calcium ion concentration in cytoplasm of hairy roots of *GmANNRJ4* transgenic soybean (**D**). Relative expression of *GmMPK6* in *GmANNRJ4* transgenic soybean plants. Results of analysis of relative expression of *GmMPK6* in *GmANNRJ4* transgenic soybean plants by qRT−PCR. Three biological replicates, each with three technical replicates; * *p* < 0.05, ** *p* < 0.01, Student’s *t* test; error bars indicate standard deviation.

**Figure 7 ijms-24-00798-f007:**
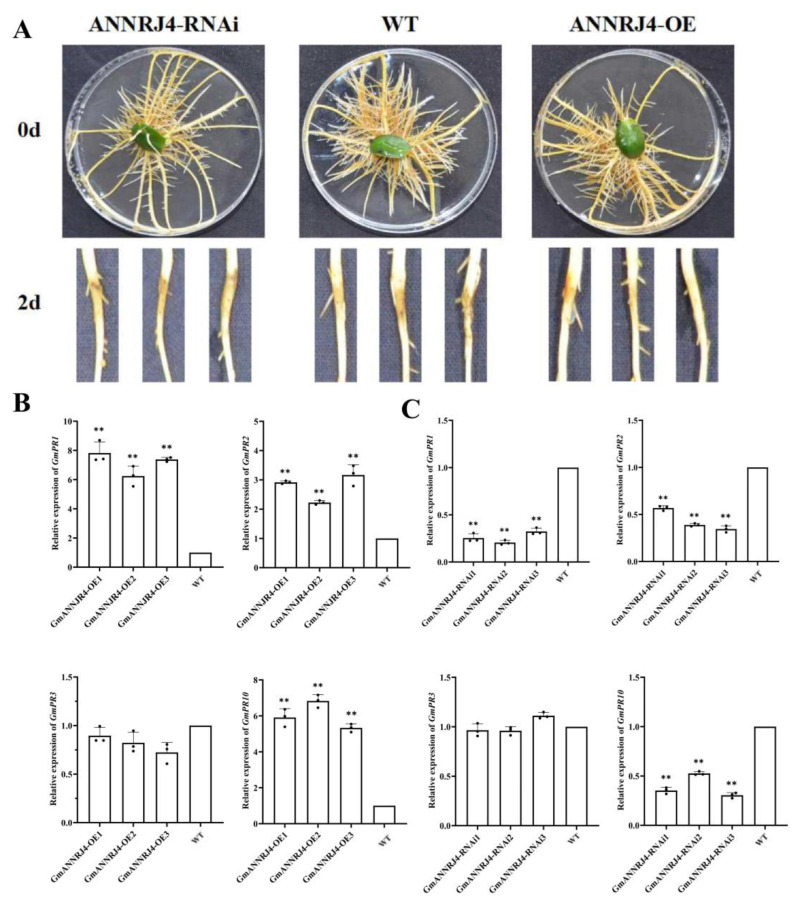
Response of *GmANNRJ4* to *P. sojae* in hairy roots of transgenic soybean. (**A**) Phenotype identification of GmANNRJ4 under *P. sojae* influence in hairy roots of transgenic soybean. (**B**,**C**) Relative expression of *GmPR1* (Glyma.15G062400), *GmPR2* (Glyma.03g132700), *GmPR3* (Glyma.02G042500), and *GmPR10* (Glyma.15G145700) in *GmANNRJ4*-OE and *GmANNRJ4*-RNAi transgenic plants. Three biological replicates, each with three technical replicates; ** *p* < 0.01, Student’s *t* test; error bars indicate standard deviation.

**Figure 8 ijms-24-00798-f008:**
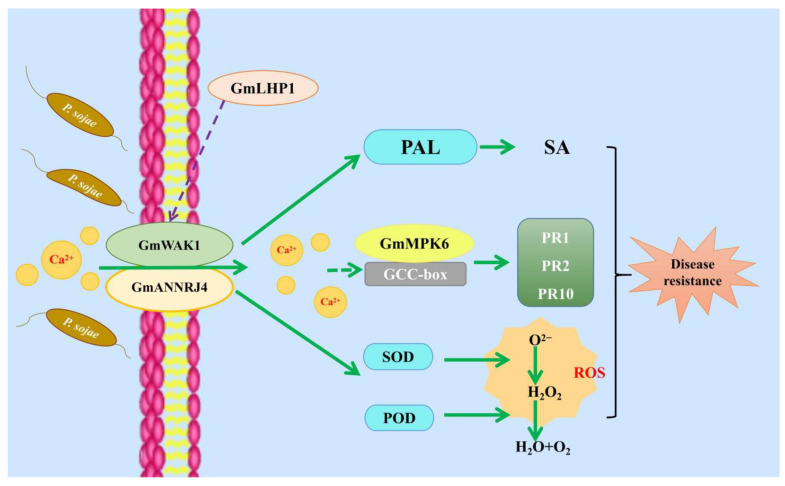
Working model illustrating how GmWAK1 and GmANNRJ4 activate defense-related genes during *P. sojae* infection.

## Data Availability

The gene accession numbers are as follows: GmWAK1 (XM_003534738), GmANNRJ4 (XM_003542786), GmSOD1 (NM_001248369), GmPOD (XM_006575142), GmActin4 (AF049106), GmEF1β (NM_001248778), GmPAL (Glyma.19G182300), GmMKK4 (Glyma_07G003200), GmMPK6 (Glyma.02G138800), GmLHP1 (Glyma_16G079900), GmPR1 (Glyma.15G062400), GmPR2 (Glyma_03g132700), GmPR3 (Glyma_02G042500), GmPR10 (Glyma.15G145700), and GmWRKY40 (Glyma.15G003300).

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
