# Peer review of "GmWAK1, Novel Wall-Associated Protein Kinase, Positively Regulates Response of Soybean to Phytophthora sojae Infection"

_ijms, 2023, doi:10.3390/ijms24010798_

Round 1

Reviewer 1 Report

Having in mind the huge economic damages worldwide caused by Phytophthora root rot on soybeans, the paper has an immeasurable contribution to the investigation of resistance mechanisms to Phytophtora. Identification of factors enhancing the soybean resistance to Phytophtora presents a new basis for soybean disease resistance molecular breeding.

Author Response

 Thanks for your comment.

Reviewer 2 Report

The study investigated the biological functions of the soybean Gmwak1 in response to P. sojae using Dongnong50 and Suinong 10’ cultivar, respectively soybean tolerant and sensitive to P. sojae. The study is very well conducted, producing a very interesting sequence of data, with concise and connected data. However, the way of presenting the data and discussing them seems to me to need to be greatly improved. Below I summarize some important points.

In title, please use the scientific rules to written Phytophthora Sojae as Phytophthora sojae

Line 21. Please present soybean as Glycine max to justify the term GmWAK1

Line 23. What does PAL mean? Every time an acronym is cited for the first time it must follow its full description. This also applies to SOD and POD. Pl, double check in the whole manuscript

Line 25. Wild type is presented in line 25, but its acronyms are presented in line 24, the order may be changed.

Line 22-26. Overexpressing (...)  decreased. 4 lines in only one sentence. Pl split this sentence into two or more sentences

Exclude Phytophthora sojae from the Keywords

Line 40. “for improving plant disease resistance” Why is this expression highlighted in the text? This goes for other parts that are also like this

Line 79. The soybean cultivar ‘Suinong 10’ is worldwide cultivated or only in China?

Lines 79-93. This large introductory text is used as a kind of summary or conclusion, while hypotheses and objectives are not presented. A light presentation of the results in the introduction is welcome, but this cannot replace the hypotheses and objectives of the work. I, therefore, suggest reducing this conclusive text and presenting the hypotheses and objectives of the work clearly.

Fig. 1. As in the International Journal of Molecular Sciences, the material and methods section is presented at the end of the manuscript, and information that can be understood in the results must be presented in the introduction. This is the case of studies with the Dongnong50 cultivar, which appears in the middle of nowhere and then disappears. I did not find other results with this cultivar. So, this data needs to be better presented, both in the introduction and in the results section.

Fig. 2. In figure 2, details are written in white font, including the caption; however, the font used is so small that the writings become practically illegible. I request that the font be used larger so that the details of the image become visible, including the scale.

Lines 182-184. “The relative expression of GmTEF1 was significantly smaller for the GmWAK1-OE lines than for the WT but significantly larger for the GmWAK1-RNAi lines (P < 0.01) (Figure 3C).” In figure 3C there are no results for GmTEF1

Lines 184-187. “After 3 days of incubation, the true leaves of WT soybean plants and GmWAK1-RNAi lines exhibited watery and even rotting lesions, while those of the GmWAK1-OE lines remained healthy (Figure 3D)”. In the 3D figure, there is no lesion result, instead relative gene expression data.

The caption of figure 3 does not include a description for image 3D, just as the 3E image lacks a description of the y-axis.

Section 2.5 should be rewritten according to the presented results. The text is confusing and does not match what is shown in figure 3.

In section 2.6, three articles are cited, which must be removed and included in the discussion section, the same happens in section 2.7

Line 225. super oxide dismutase may be written as superoxide dismutase

In the figure 4 subtitle, The activity of PAL and the expression of relative gene GmPAL in leaves of transgenic and wild-type should be written as The expression of relative gene GmPAL (B) and activity of PAL and the in leaves of transgenic and wild-type (C).

Figures 5A and 5B are the same figure and do not contain POD data. Furthermore, the legend should be written as The activity of SOD (A), POD (B) and ROS (C), and H2O2 (D) accumulation. As it stands, it is evident that figure A would show POD activity data, as well as figure B data on SOD activity. They would likewise show H2O2 ROS activity when it should be accumulation, not activity.

Lines 228-229. “under both mock treatment and at 24 hpi (Figure 5A, 5B).” Very confused, please rewritten it. What is 24 hpi? The same as 24 hours??

Line 231. ROS activity??? What do the authors mean by this? Is ROS an enzyme or an accumulated metabolite after oxidative stress events? The same error is done in Fig. 5, inducing the reader to think that ROS is an enzyme, as well as H2O2. Many times in this manuscript the mistake as this is verified. Pl, double check

Line 236-239. “Accordingly, we deduced that the antioxidant enzymatic activities were increased because of the higher expression level of the corresponding enzymatic genes in GmWAK1 soybean plants, and thus may eliminate ROS to alleviate the harm from P. sojae infection.” This statement is physiologically wrong, as the increased activity of SOD and POD generally eliminates O2- but produces H2O2 which is even more reactive than O2-.

Line 258. Bimolecular or Biomolecular?

In the text as a whole, there are already several abbreviations that must appear in full in the first appearance and after this only the abbreviation can appear, but in many of the occurrences, there is no description of the abbreviation. Pl, double check

Lines 286-435. In this great text, the authors redescribe what is already well described in the literature. I believe this is unnecessary, so I suggest that this text be reduced by half or a third.

Tables and figures are an independent part of the text. Therefore, they must be clear and complete to the point of not need to return to the text to understand them. However, the tables and graphs presented are completely devoid of details and with symbology that is not shown in the figures or in the text. This part of the manuscript deserves greater attention.

The scheme in figure 8 should be improved, as it makes it clear that the responses activated by P. soyae induce the expression or greater activity (this is not clear in the scheme) of SOD and POD and these lead to a reduction in ROS production. This last information is wrong, since both SOD and POD use superoxide anions, producing hydrogen peroxide, a molecule that, if not metabolized quickly, is more harmful than the superoxide anion, therefore the continuation of the reaction must be presented until the cleavage of H2O2 into H2O + ½ O2.

Line 444. The amount of light is presented in LUX, a unit that was widely used in the 19th and mid-20th centuries. This unit of measurement of the light intensity has been updated to PAR (photosynthetic active radiation) measured in mol m-2 s-1. Please do the conversion from LUX to PAR.

Line 452-453. The principle of scientific experimentation is to allow the reproduction of experiments published by a team in another team. For this, the methodological description must follow very clear principles, a fact that is presented here very succinctly. For example, we have Trizol reagent which is labeled as Invitrogen. However, on the Invitrogen website, there are several types of Trizol, each with its Part Number. In this sense, this and all other reagents must be written in full, showing the manufacturer and country of manufacture, as well as the Part Number. Take as an example Parafilm® M which is manufactured by MilliporeSigma, Merck, Massachusetts, USA with a Part Number of P7543.

Lines 602-603. What do the authors mean by ROS and H2O2 activity levels? Activity is more used for enzyme-promoted metabolic activity and level is more used for accumulation; therefore, the authors have to be clear in informing what exactly is being presented. This confusion is noticed in several parts of the manuscript, which should be double-checked.

Author Response

  1. In title, please use the scientific rules to written Phytophthora Sojae as Phytophthora sojae

Response: Thanks for your comment. We have changed the title as ‘GmWAK1, a Novel Wall-Associated Protein Kinase, Positively Regulates the Response of Soybean to Phytophthora sojae Infection’(Please see page 1 line 2).

  1. Line 21. Please present soybean as Glycine max to justify the term GmWAK1

 Response: Thanks for your comment. We have changed the soybean to Glycine max (Please see page 2 line 28).

  1. Line 23. What does PAL mean? Every time an acronym is cited for the first time it must follow its full description. This also applies to SOD and POD. Pl, double check in the whole manuscript

 Response: Thank you for your advice. We have written the full name for PAL, SOD,POD and ROS for the first time: Phenylalanine ammonia-lyase (PAL), superoxide dismutase (SOD), peroxidase (POD), wild-type (WT), reactive oxygen species (ROS), hydrogen peroxide (H2O2)

and pathogenesis-related (PR). (Please see page 2 ).

  1. Line 25. Wild type is presented in line 25, but its acronyms are presented in line 24, the order may be changed.

 Response: Thank you for your advice. We have changed the order (Please see page 2 line 33).

  1. Line 22-26. Overexpressing (...)  decreased. 4 lines in only one sentence. Pl split this sentence into two or more sentences.

Response: Thank you for your advice. We have rewritten this sentence as‘  Overexpression of GmWAK1 in soybean significantly improved resistance to P. sojae, and the levels of phenylalanine ammonia-lyase (PAL), SA, and SA-biosynthesis-related genes were markedly higher than in the wild-type (WT) soybean. The activities of enzymatic superoxide dismutase (SOD) and peroxidase (POD) antioxidants in GmWAK1-overexpressing (OE) plants were significantly higher than those in in WT plants treated with P. sojae; reactive oxygen species (ROS) and hydrogen peroxide (H2O2) accumulation was considerably lower in GmWAK1-OE after P. sojae infection. ’(Please see page 2 line 31-38).

  1. Exclude Phytophthora sojae from the Keywords

  Response: Thanks for your comment. We have Excluded Phytophthora sojae from the Keywords. (Please see page 3 line 46).

  1. Line 40. “for improving plant disease resistance” Why is this expression highlighted in the text? This goes for other parts that are also like this

Response: Thank you for your advice. We have checked and revised the font, size, space etc. in the full manuscript to make it consistent. The font size of “for improving plant disease resistance” have been revised.(Please see page 3 line 60).

  1. Line 79. The soybean cultivar ‘Suinong 10’ is worldwide cultivated or only in China?

 Response: Thank you for your advice. ‘Suinong 10’ is cultivated only in China. ‘Suinong 10’ is resistant to the predominant race of P. sojae race 1 in Heilongjiang,

  1. Lines 79-93. This large introductory text is used as a kind of summary or conclusion, while hypotheses and objectives are not presented. A light presentation of the results in the introduction is welcome, but this cannot replace the hypotheses and objectives of the work. I, therefore, suggest reducing this conclusive text and presenting the hypotheses and objectives of the work clearly.

 Response: Thank you for your constructive comment. We have added the purpose of the study at the end of the introduction as follows: As such, our aims in the current study included (a) using expression analysis and the phenotype to verify whether GmWAK1 is involved in the response to P. sojae infection; (b) finding the protein that interacts with GmWAK1 through a yeast two-hybrid assay and verifying its functionality; and (c) examining the physiological indicators and SA content to determine the molecular mechanism through which GmWAK1 responds to P. sojae. Our results provide a new perspective for in-depth analysis of the soybean response to P. sojae infection. (Please see page 5 line 96).

  1. 1. As in the International Journal of Molecular Sciences, the material and methods section is presented at the end of the manuscript, and information that can be understood in the results must be presented in the introduction. This is the case of studies with the Dongnong50 cultivar, which appears in the middle of nowhere and then disappears. I did not find other results with this cultivar. So, this data needs to be better presented, both in the introduction and in the results section.

 Response: Thank you for your constructive comment. We have added this part at the the introduction as follows: Soybean cultivars ‘Dongnong 50’ and ‘Suinong 10’ are resistant and susceptible, respectively, to the predominant race1 of P. sojae  in Heilongjiang, China [5]; these materials are valuable for the study of plant resistance to P. sojae. (Please see page 3 line 53).

  1. 2. In figure 2, details are written in white font, including the caption; however, the font used is so small that the writings become practically illegible. I request that the font be used larger so that the details of the image become visible, including the scale.

 Response: Thank you for your advice. In figure 2, details written in white font are the scale. We have enlarged the front in figure 2.

  1. Lines 182-184. “The relative expression of GmTEF1 was significantly smaller for the GmWAK1-OE lines than for the WT but significantly larger for the GmWAK1-RNAi lines (P < 0.01) (Figure 3C).” In figure 3C there are no results for GmTEF1. Lines 184-187. “After 3 days of incubation, the true leaves of WT soybean plants and GmWAK1-RNAi lines exhibited watery and even rotting lesions, while those of the GmWAK1-OE lines remained healthy (Figure 3D)”. In the 3D figure, there is no lesion result, instead relative gene expression data. The caption of figure 3 does not include a description for image 3D, just as the 3E image lacks a description of the y-axis. Section 2.5 should be rewritten according to the presented results. The text is confusing and does not match what is shown in figure 3.

 Response: Thank you for your advice. We have written Section 2.5 as ‘In the GmWAK1-RNAi lines, the cotyledons inoculated with P. sojae exhibited water-soaked lesions and were softer than those of WT; we observed almost no disease symptoms in the GmWAK1-OE lines. The P. sojae lesions on the GmWAK1-OE plants were significantly smaller than those on WT plants (P < 0.01), but larger than those on the GmWAK1-RNAi plants (Figure 3A). We obtained similar results after the inoculation of true leaves with P. sojae. After 3 days of incubation, the true leaves of WT and GmWAK1-RNAi transgenic soybean plants exhibited watery and rotting lesions, whereas those of the GmWAK1-OE transgenic soybean plants remained healthy (Figure 3B). As observed for the cotyledons, the lesion areas were significantly smaller in the GmWAK1-OE transgenic soybean plants and significantly larger in the GmWAK1-RNAi transgenic soybean plants relative to the WT plants (Figure 3C). The relative expression of GmTEF1 was significantly lower in in cotyledons and true leaves of GmWAK1-OE lines than in the WT plants but significantly higher in the GmWAK1-RNAi lines (P < 0.01) (Figure 3D and 3E). These results suggested that when GmWAK1 is overexpressed, the transgenic soybean plants acquired higher resistance to P. sojae infection. (Please see page 7 line 159-174)

  1. In section 2.6, three articles are cited, which must be removed and included in the discussion section, the same happens in section 2.7.

Response: Thank you for your advice. We have removed this sentences and added it in the discussion section.

  1. Line 225. super oxide dismutase may be written as superoxide dismutase

Response: Thank you for your advice. We have changed it to superoxide dismutase. (Please see page 9 line 192).

  1. In the figure 4 subtitle, The activity of PAL and the expression of relative gene GmPAL in leaves of transgenic and wild-type should be written as The expression of relative gene GmPAL (B) and activity of PAL and the in leaves of transgenic and wild-type (C).

Response: Thank you for your advice. We have rewritten as (B) The Phenylalanine ammonia-lyase (PAL) activity in transgenic soybean plants and mock-treated WT plants at 0 and 24 h after P. sojae infection. (C) Relative expression GmPAL gene in leaves of transgenic and WT plants. (Please see page 42 line 907).

  1. Figures 5A and 5B are the same figure and do not contain POD data. Furthermore, the legend should be written as The activity of SOD (A), POD (B) and ROS (C), and H2O2 (D) accumulation. As it stands, it is evident that figure A would show POD activity data, as well as figure B data on SOD activity. They would likewise show H2O2 ROS activity when it should be accumulation, not activity.

Response: Thank you for your advice. We have changed the legend of Figure5 as Activities of superoxide dismutase (SOD) (A), peroxidase (POD) (B), and reactive oxygen species (ROS) (C), hydrogen peroxide (H2O2) (D) accumulation.(Please see page 42 line 913) We have checked the Figure5, and changed it to accumulation.

  1. Lines 228-229. “under both mock treatment and at 24 hpi (Figure 5A, 5B).” Very confused, please rewritten it. What is 24 hpi? The same as 24 hours??

Response: Thank you for your advice. 24 hpi means 24 h postinoculation (hpi), it was first appeared Section 2. 6 page 9 Line 184.

  1. Line 231. ROS activity??? What do the authors mean by this? Is ROS an enzyme or an accumulated metabolite after oxidative stress events? The same error is done in Fig. 5, inducing the reader to think that ROS is an enzyme, as well as H2O2. Many times in this manuscript the mistake as this is verified. Pl, double check.

Response: Thank you for your advice. We have checked all the manuscript, changed the activity as  accumulation. (Please see page 9 line 197)

  1. Line 236-239. “Accordingly, we deduced that the antioxidant enzymatic activities were increased because of the higher expression level of the corresponding enzymatic genes in GmWAK1 soybean plants, and thus may eliminate ROS to alleviate the harm from sojae infection.” This statement is physiologically wrong, as the increased activity of SOD and POD generally eliminates O2-but produces H2O2which is even more reactive than O2-.

Response: Thank you for your advice. We have changed as ‘According to the results, we inferred that due to the elevated expression level of the corresponding enzyme genes in GmWAK1 transgenic soybean plants, the antioxidant enzyme activity increased, which may to keep maintain at a lower certain steady-state levels of ROS, preventing harm from P. sojae infection.’(Please see page 10 line 201). And in past studies showed that , in plant , O2- can be further reduced by SOD dismutation to H2O2, meanwhile, Plant peroxidases (POD) reduce hydrogen peroxide (H2O2) in the presence of an electron donor[76, 77]. So we thought that the H2O2 which coused by SOD could reduced by the peroxidase (POD) and other Enzymes, to maintain at a lower certain steady-state levels of ROS [78,79], to alleviate the harm from Pathogens infection. So, we thought when the activities of SOD and POD were increased, that may could keep a the balance of ROS in a lower level. Thanks again.

  1. Line 258. Bimolecular or Biomolecular?

Response: Thank you for your advice. BiFC means Bimolecular fluorescence complementation. So it should be Bimolecular.

  1. In the text as a whole, there are already several abbreviations that must appear in full in the first appearance and after this only the abbreviation can appear, but in many of the occurrences, there is no description of the abbreviation. Pl, double check

Response: Thank you for your advice. We have checked all the text , written the full name before the abbreviation of Phytophthora sojae (P. sojae)  Wall-associated protein kinase (WAK) receptor-like protein kinase (RLK) salicylic acid (SA) overexpression (OE) reactive oxygen species (ROS) hydrogen peroxide (H2O2) pathogenesis-related (PR)  (Please see page 2 line 25, 26, 31, 35,36, 41). Quantitative reverse transcription-polymerase chain reaction (qRT-PCR) (Please see page 6 line 130), green fluorescent protein (GFP)(Please see page 7 line 145) bimolecular fluorescence complementation (BiFC) (Please see page 10 line 121)

  1. Lines 286-435. In this great text, the authors redescribe what is already well described in the literature. I believe this is unnecessary, so I suggest that this text be reduced by half or a third.

Response: Thank you for your advice. We have reduced some redundant content in this section.

  1. Tables and figures are an independent part of the text. Therefore, they must be clear and complete to the point of not need to return to the text to understand them. However, the tables and graphs presented are completely devoid of details and with symbology that is not shown in the figures or in the text. This part of the manuscript deserves greater attention.

Response: Thank you for your advice. We have cheked and changed all the tables and figures and figure legends.

  1. Line 444. The amount of light is presented in LUX, a unit that was widely used in the 19th and mid-20th centuries. This unit of measurement of the light intensity has been updated to PAR (photosynthetic active radiation) measured in mol m-2 s-1. Please do the conversion from LUX to PAR.

Response: Thank you for your advice. We have chenged the amount of light to 110 PAR. (Please see page 17 line 372).

  1. The scheme in figure 8 should be improved, as it makes it clear that the responses activated by P. soyae induce the expression or greater activity (this is not clear in the scheme) of SOD and POD and these lead to a reduction in ROS production. This last information is wrong, since both SOD and POD use superoxide anions, producing hydrogen peroxide, a molecule that, if not metabolized quickly, is more harmful than the superoxide anion, therefore the continuation of the reaction must be presented until the cleavage of H2O2into H2O + ½ O2.

Response: Thank you for your advice. We have rewrriten this part as ‘when soybean is infected by P. sojae, the expression of GmWAK1 increases, the expression of endogenous SA is upregulated, and the activities of SOD and POD are upregulated, which reduce the accumulation of O2- and H2O2, to maintain at a lower certain steady-state levels of ROS;’ (Please see page 17 line 354). And we have changed the Figure8.

  1. Line 452-453. The principle of scientific experimentation is to allow the reproduction of experiments published by a team in another team. For this, the methodological description must follow very clear principles, a fact that is presented here very succinctly. For example, we have Trizol reagent which is labeled as Invitrogen. However, on the Invitrogen website, there are several types of Trizol, each with its Part Number. In this sense, this and all other reagents must be written in full, showing the manufacturer and country of manufacture, as well as the Part Number. Take as an example Parafilm® M which is manufactured by MilliporeSigma, Merck, Massachusetts, USA with a Part Number of P7543.

Response: Thank you for your advice. We have added all the Part Number in the manuscript:TRIzol® Reageant with a Part Number of 15596026. MLV reverse transcriptase kit (Takara, Dalian, China, with a Part Number of 2641A). pEASY® Blunt vector (Transgen Biotech, Beijing, China, with a Part Number of CB101(Please see page 18 line 379, 381, 383). real-time PCR kit (Toyobo, Japan, with a Part Number of FSQ-201) (Please see page 18 line 392). anti-His antibody and anti-GST antibody (Abmart, United States with a Part Number of M20020 and M20025). (Please see page 22 line 470)

  1. Lines 602-603. What do the authors mean by ROS and H2O2 activity levels? Activity is more used for enzyme-promoted metabolic activity and level is more used for accumulation; therefore, the authors have to be clear in informing what exactly is being presented. This confusion is noticed in several parts of the manuscript, which should be double-checked

Response: Thank you for your advice. We have checked all the manuscript, changed the activity as  accumulation. (Please see page 23 line 497)

Reviewer 3 Report

Manuscript entitled “GmWAK1, a Novel Wall-Associated Protein Kinase, Positively Regulates the Response of Soybean to Phytophthora Sojae In-fection” having significance in the area of specificity.

Experiment has significant finding that might be useful in the future. Below is the suggestion that might be helpful in the improvement of manuscript.

Abstract

In this section some significant finding with values should be mentioned.

Additional section needs English improvements

Introduction

First paragraph requires English and grammar Improvements

Line 40- maintain font size

Introduction section should contain significant of crop and the loss caused worldwide to the productivity of this crop due to this pathogen.

Results

Line 99- should be shifted to discussion section

Quality of figures needs improvements; font size has not been made uniform, it should be checked for all figures

Line 148-149- font is not uniform

Some section of results seems part of discussion; it should be check through the entire result section.

Line 201- should be shifted to discussion section

Line 224-226- why in result section

Discussion

Needs English improvements

Line 352-355- font is not uniform

Discussion has been written fine, I suggests authors should avoid mentioning figure number in discussion section as it has already presented in the result section.

Methodology section needs English and grammar Improvements.

References:

Check entire references for uniform formatting

Can be considered after minor revision

Author Response

  1. Abstract: In this section some significant finding with values should be mentioned. Additional section needs English improvements

Response: Thanks for your advice. We have rewritten the Abstract. We have undergone English language editing by MDPI. The text has been checked for correct use of grammar and common technical terms, and edited to a level suitable for reporting research in a scholarly journal.

  1. Introduction: First paragraph requires English and grammar Improvements. Line 40- maintain font size. Introduction section should contain significant of crop and the loss caused worldwide to the productivity of this crop due to this pathogen.

Response: Thanks for your advice. The font size of “for improving plant disease resistance” have been revised. (Please see page 3 line 60). We have add the introduction as follow Phytophthora root rot is a kind of destructive and worldwide disease in soybean which caused by the oomycete pathogen P. sojae [1, 2]. It has a rapid incidence and wide spread, which could result in serious economic losses [3, 4]. So far, using genetic resistance in soybean has been one of the most effective approachs to reduce losses caused by P. sojae. (Please see page 3 line 49). The text has undergone English language editing by MDPI.

  1. Results: Line 99- should be shifted to discussion section.

Response: Thanks for your advice. We have removed this sentences and added it in the discussion section.

  1. Line 148-149- font is not uniform.Some section of results seems part of discussion; it should be check through the entire result section.

Response: Thanks for your advice. The font size of ‘These results suggested that GmWAK1 was primarily involved in the response to P. sojae and SA treatment.’ have been revised. (Please see page 7 line 143). We have removed this sentences and added it in the discussion section.

  1. Line 201- should be shifted to discussion section.

Response: Thanks for your advice. We have removed this sentences and added it in the discussion section.

  1. Line 224-226- why in result section.

Response: Thanks for your advice. We have removed this sentences and added it in the discussion section.

  1. Quality of figures needs improvements; font size has not been made uniform, it should be checked for all figures.

Response: Thanks for your advice. We have checked and improved all the figures, and made the uniform font size.

  1. Discussion: Needs English improvements.

Response: Thanks for your advice. The text has undergone English language editing by MDPI.

  1. Line 352-355- font is not uniform.

Response: Thanks for your comment. The font size of ‘ Annexins form permeable calcium channels on cell membranes and play a major role in regulating the concentrations of reactive oxygen species and free Ca2+ in plant cells [49,50,54]. ’ have been revised. (Please see page 14 line 292)

  1. Discussion has been written fine, I suggests authors should avoid mentioning figure number in discussion section as it has already presented in the result section.

Response: Thanks for your comment. we have deleted all the figure number in discussion section.

  1. Methodology section needs English and grammar Improvements.

Response: Thanks for your comment. The text has undergone English language editing by MDPI.

  1. References: Check entire references for uniform formatting

Response: Thanks for your comment. We have checked and rewritten all the references to uniform formatting (Please see page 24 to page 40 ).
